solid state physics

memristor, mobile vacancies, Burgers' equation, switching time

**Author for correspondence:**
I. V. Boylo
e-mail: boylo@fti.dn.ua

# Nonlinear effects in memristors with mobile vacancies

I. V. Boylo[1] and K. L. Metlov[1,2]

[1]Donetsk Institute for Physics and Technology, R. Luxembourg str. 72, 83114 Donetsk, Ukraine
[2]Institute for Numerical Mathematics RAS, 8 Gubkina str., 119991 Moscow GSP-1, Russia

 IVB, 0000-0003-1543-9764; KLM, 0000-0002-3929-5665

Because local concentration of vacancies in any material is bounded, their motion must be accompanied by nonlinear effects. Here, we look for such effects in a simple model for electric field-driven vacancy motion in a memristor, solving the corresponding nonlinear Burgers' equation with impermeable nonlinear boundary conditions exactly. We find non-monotonous relaxation of the resistance while switching between the stable ('on' and 'off') states, and qualitatively different dependencies of switching time (under applied current) and relaxation time (under no current) on the memristor length. Our solution can serve as a useful benchmark for simulations of more complex memristor models.

## 1. Introduction

Memristors were proposed by Chua [1] as another (originally missing) building block for electric circuits. Their main feature is hysteresis in the current–voltage characteristic or the possibility of having (and switching between) different resistive states. Nowadays, memristors have become an important part of developing information storage techniques, such as ReRAM [2], in-memory computing [3], neuromorphic computations [4,5] and other applications [6].

Among different types of memristors, the ones in which oxygen vacancy movement processes play the key role in formation of resistive states have attracted substantial attention [7–10]. Such states are formed due to much higher mobility of vacancies than that of the metal cations in many transition metal oxides [11].

A model for these states can be formulated in terms of the position of the interface between the vacancy-rich and the vacancy-depleted regions [2] moved by the electric field. It can be further generalized by describing its input–output relationship using Bernoulli ordinary differential equation [12]. A more detailed description in terms of spatial distribution of the vacancy concentration was developed in many recent modelling works [13–21], where the underlying kinetic equations were solved numerically. Inherent nonlinearity in these models can be expressed via the nonlinear diffusion equation, predicting a

prominent nonlinear effect—formation of vacancy concentration shock waves [22]. The purpose of this article is to look for other essentially nonlinear effects (absent in the linear approximation) in the nonlinear vacancy diffusion-induced switching behaviour of memristors. Here, we study a simpler (but exactly solvable) memristor model in strongly nonlinear regime, reducing to Burgers' equation for vacancy concentration with impermeable nonlinear boundary conditions, which is formulated next.

## 2. Exactly solvable memristor

Consider a thin film of a material (with charged mobile vacancies) sandwiched between two metals. The coordinate $x$ is counted in the direction perpendicular to the film, which has thickness $d$. Assume that memristive interfaces at $x = 0$ and $x = d$ are impermeable for vacancies, so that their total number in the film is conserved. The state of such a memristor at a time $t$ is described by the instantaneous local vacancy concentration $C(x, t)$. Because the number of vacancies is also locally conserved, $C(x, t)$ obeys the continuity equation

$$\partial_t C(t, x) + \nabla \cdot \mathbf{J}(t, x) = 0, \tag{2.1}$$

where $\partial_t$ is the time derivative, and $\nabla \cdot \mathbf{J}(t, x) = \partial_x J(t, x)$ in the considered one-dimensional context when $\mathbf{J} = \{J, 0, 0\}$. Suppose that each of the vacancies lives in a periodic potential with the distance $a$ between its minima, separated by energy barriers of the height $U_A$. Interaction of the vacancy electric charge $q$ with the local electric field $E$ makes the potential skewed, setting a preferred direction for vacancy jumps. The probability to overcome the energy barrier and move forward → or backward ← into the neighbouring energy minimum [23] can be expressed as follows:

$$r_{\rightleftarrows} = \frac{\nu}{2} \exp\left(-\frac{U_A \mp aqE}{k_B T}\right), \tag{2.2}$$

where $\nu$ is the attempt frequency, $k_B$ is the Boltzmann constant and $T$ is the absolute temperature. From Ohm's Law $E = \rho_0 I$, where the resistivity $\rho_0$ is assumed here to be a constant and $I$ is the electric current density.

The vacancies can only make a jump if (i) they are present at the original energy minimum and (ii) there is free space for them (e.g. a movable oxygen atom in the case of oxygen vacancies) at the location of neighbouring energy minimum. The corresponding joint probability is $c(1 - c)$, where the normalized mobile vacancy concentration $c = (C - C_{min})/(C_{max} - C_{min})$ is defined assuming that $C_{min} \leq C \leq C_{max}$, so that $0 \leq c \leq 1$. The value of $C_{min}$ is the concentration of immobile vacancies and $C_{max}$ is the maximum concentration of vacancies, determined by the chemical composition of the film's material.

Summarizing, we can express the vacancy drift current due to the electric current $I$ as follows:

$$J_{drift} = c(1 - c)a(r_{\rightarrow} - r_{\leftarrow})$$
$$= c(1 - c)\frac{2D}{a}\sinh\frac{aq\rho_0 I}{k_B T}, \tag{2.3}$$

where $D = a^2\nu/2\exp(-U_A/(k_B T))$ is the diffusivity. Another contribution to the vacancy current is due to diffusion and can be described by the Fick's Law $J_{diff} = -D\partial_x c$. Substituting the total current $J = J_{drift} + J_{diff}$ into (2.1) and renormalizing the coordinate $\xi = x/d$, time $\tau = tD/d^2$ and the vacancy current $j = Jd/D$, we arrive at the nonlinear Burgers' equation for the dimensionless vacancy concentration $c(\tau, \xi)$ [22]

$$\partial_\tau c + p(1 - 2c)\partial_\xi c = \partial_{\xi\xi}c, \tag{2.4}$$

where $p = 2(d/a)\sinh(a\,q\rho_0 I/(k_B T)) = const$. It reduces to the canonical Burgers' equation for the function $\widetilde{c} = 1 - 2c$, but we will solve it directly for $c$ in the present form. It is interesting that the external force (electric field $\rho_0 I$) enters equation (2.4) via a coefficient before the nonlinear term, but not as a separate term on the right-hand side like in many classical equations of mathematical physics.

To solve equation (2.4), first introduce the antiderivative function

$$u(\tau, \xi) = \int_0^\xi c(\tau, \zeta)\,\mathrm{d}\zeta, \tag{2.5}$$

noting that $u(\tau, 1) = r$ is the total number of vacancies per unit of the film area (filling ratio). Integrating (2.4) over $\xi$ produces the equation for $u$: $\partial_\tau u + p(\partial_\xi u - (\partial_\xi u)^2) = \partial_{\xi\xi}u$. After the substitution $u = (1/p)\log h$ (known as the Hopf–Cole substitution [24,25] or the Molenbroek–Chaplygin hodograph method [26]), nonlinear terms in the equation for $h(\tau, \xi)$ are cancelled and we recover the linear diffusion equation:

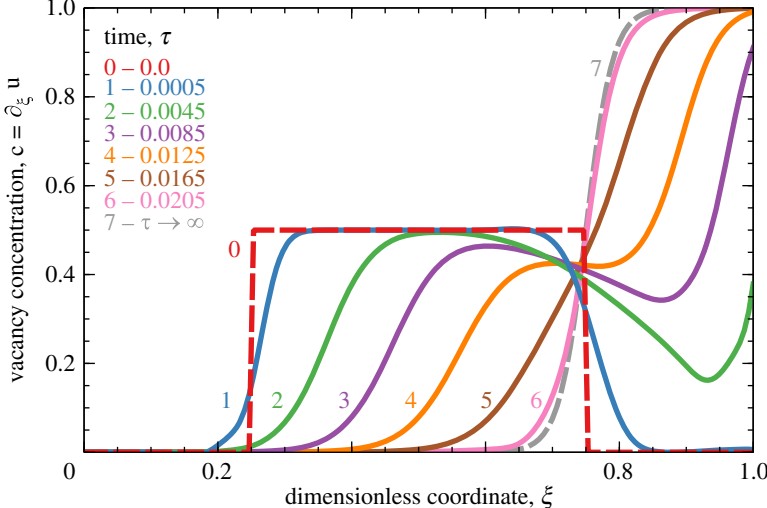

**Figure 1.** Evolution of the square bump (shown by the dashed line at $\tau = 0$) in vacancy concentration under influence of the current flowing in the positive direction, computed from (2.6) with (3.1) and $p = 50$.

$\partial_\tau h + p\,\partial_\xi h = \partial_{\xi\xi}h$. Its solution can be represented as a sum of a particular solution (satisfying inhomogeneous boundary conditions) and the general solution for the homogeneous boundary conditions of the corresponding type. Because the number of vacancies in the film is conserved, $u(\tau, 0) = 0$ and $u(\tau, 1) = r$, the boundary conditions are of Dirichlet type: $h(\tau, 0) = 1$ and $h(\tau, 1) = e^{pr}$. Guessing the particular solution $P$ and finding the general solution for the problem with homogeneous boundary conditions by separation of variables we get

$$u = \frac{\log\left[P + p\,e^{-\tau p^2/4 + p\xi/2}\sum_{n=1}^{\infty}h_n\,e^{-\tau n^2\pi^2}\sin n\pi\xi\right]}{p}$$

and

$$P(\xi; p, r) = \frac{e^p - e^{pr} - e^{p\xi} + e^{p(r+\xi)}}{e^p - 1},$$

(2.6)

where the semicolon in function arguments separates (sometimes omitted) constant parameters $p$ and $r$. One can verify directly that for any set of the Fourier coefficients $h_n$, the corresponding $c = \partial_\xi u$ satisfies the original Burgers' equation (2.4) and has exactly zero vacancy current $j = p\,c\,(1 - c) - \partial_\xi c$ at the boundaries: $j(\tau, 0) = j(\tau, 1) = 0$ at all times. This exact solution depends on the total number of vacancies in the film $r$.

The coefficients $h_n$ and the value of $r$ can be uniquely determined from the initial conditions. Given $c(0, \xi) = c_0(\xi)$, we can compute $u(0, \xi) = u_0(\xi) = \int_0^\xi c_0(\xi)\,d\xi$ and using orthonormality of $\sin n\pi\xi$ obtain

$$h_n = 2\int_0^1 \frac{e^{pu_0(\xi)} - P(\xi; p, r)}{p}\,e^{-p\xi/2}\sin n\pi\xi\,d\xi,$$

(2.7)

with $r = u_0(1)$. It is worth noting that under such definition of Fourier coefficients $h_n$, they attain finite limit at $p \to 0$ and consequently $P \to 1$. In this limit, the derivative $c = \partial_\xi u$ of (2.6) becomes the cosine series solution of the Cauchy problem for the linear (with $p = 0$) diffusion equation (2.4) with homogeneous linear Neumann-type boundary conditions (no vacancy current $j = j_{\text{diff}} = 0$ at either boundary), which relaxes into the uniform state $\lim_{\tau\to\infty}c(\tau, \xi; 0, r) = r$.

## 3. Driven vacancy motion

For example, take initial distribution of vacancies in the form of rectangular bump $c_0^{\text{rect}}(\xi) = [\theta(\xi - 1/4) - \theta(\xi - 3/4)]/2$ so that $u_0^{\text{rect}}$ is a piecewise linear function, $u_0^{\text{rect}}(1) = r^{\text{rect}} = 1/4$ and

$$h_n^{\text{rect}} = e^{-p/8}\frac{\sin(n\pi/4)}{n\pi/4}\,\text{Im}\,\frac{e^{in\pi/2}}{p - 2\imath\pi n},$$

(3.1)

where $\imath = \sqrt{-1}$ and Im denotes taking imaginary part of a complex number. Substituting (3.1) into (2.6), we obtain evolution of this initial bump plotted in figure 1. As one can see, the external field causes the initial bump to relax into a stable configuration at $\tau \to \infty$. The analytical expression for this configuration

follows from (2.6): $c_{st}(\xi; p, r) = (1/p)\partial_\xi \log P$, which depends on the current $p$ and the total number of vacancies in the initial state $r = u_0(1)$. This stable profile forms as the result of the competition between the directed jumps due to the applied current, trying to push the vacancies against the boundary, and undirected jumps due to the diffusion, trying to even out the vacancy distribution. Except for the value of $r$, all information about the initial conditions is lost in the final state.

For a given magnitude of $|p|$, there are two distinct stable configurations for positive $p > 0$ and negative $p < 0$ current directions. We will call them 'on' and 'off' states, respectively: $c_{on} = c_{st}(\xi; |p|, r)$ and $c_{off} = c_{st}(\xi; -|p|, r)$. These states are mirror-symmetric: $c_{off}(\xi; |p|, r) = c_{on}(1 - \xi; |p|, r)$ and $P(1 - \xi; p, r) = e^{pr} P(\xi; -p, r)$. Switching between them is the main operating mode of the memristor.

To consider the switching, it is necessary to express the $c_{off}$ state in the Fourier basis, corresponding to the positive direction of the current $p > 0$. By using (2.7), we get

$$
\left.\begin{aligned}
h_n^{off} &= \frac{1}{p}\left( S(n, p, r) - \frac{8n\pi(1 - (-1)^n e^{p(r-1/2)})}{p^2 + 4n^2\pi^2} \right), \\
S &= 2\int_0^1 e^{-(p\xi/2)} \frac{e^{pr}}{P(1 - \xi; p, r)} \sin n\pi\xi \, d\xi \\
&= \frac{2(-1)^{n+1} e^{-p/2}(1 + g\,e^p)}{g\,p} \operatorname{Im}\left[(-g)^{\frac{1}{2}+(in\pi/p)}Q\right]
\end{aligned}\right\}
$$

and
$$
Q = B\left(-g\,e^p, \frac{1}{2} - \frac{in\pi}{p}, 0\right) - B\left(-g, \frac{1}{2} - \frac{in\pi}{p}, 0\right),
$$
(3.2)

where $g = (e^{pr} - 1)/(e^p - e^{pr}) > 0$ and $B(z, a, b) = \int_0^z z^{a-1}(1 - z)^{b-1}\,dz$ is the incomplete Euler's beta function. Similar to (3.1), the second expression for $S$ was obtained by extending the integrand into the complex plane and making the substitution $\xi = -(2/p)\log\eta$, which turns it into a product of a rational function in $\eta$ and a power $\eta^\alpha$ that can be rewritten in terms of beta functions. Equations (3.2) and (2.6) make it possible to compute the evolution of the memristor state during the switching. We have used a finite number (3000) of terms in the infinite series (2.6), which is sufficient for plotting the subsequent figures.

# 4. Evolution and relaxation of the resistance

Since it was assumed from the start that the local resistivity $\rho_0$ is constant (which is the main leading order contribution), the change of the resistance in the present model may only come from the interfaces

$$
\rho(x) = \rho_0 + \frac{\kappa_1(x)\delta(x/d)}{d} + \frac{\kappa_2(x)\delta(x/d - 1)}{d},
$$
(4.1)

where $\kappa_i$ stands for the surface resistivity of the interface $i$ and Dirac's delta functions $\delta(x)$ are assumed to be left handed, sitting just outside the range $x \in [0, d]$. This resistance model corresponds to the case (like in some transition metal oxides [14,16]), when the bulk resistance change is negligible compared to the interfacial one. The opposite situation may also take place [27,28], but it is beyond the scope of the present analytical treatment.

There are several possible resistivity change mechanisms (disruption of the crystal structure by defects, valence change of the Mn ions, formation of Schottky barrier, etc.). Without delving into details, let us assume a phenomenological series expansion

$$
\kappa_i(x) = \kappa_{i,0} + k_{i,1}c(x),
$$
(4.2)

where the constants $\kappa_{i,0}$ and $k_{i,1}$ are defined by the material of the contact $i$, the material of the main body of the memristor (including its immobile vacancies) and the contact type. Integrating (4.1) from $x = 0 - 0$ to $x = d + 0$ (covering the delta functions), we get for the total resistance

$$
R = R_0 + k_1 c(0) + k_2 c(1),
$$
(4.3)

where $R_0 = (d\rho_0 + \kappa_{1,0} + \kappa_{2,0})/A$, $k_1 = k_{1,1}/A$, $k_2 = k_{2,1}/A$ and $A$ is the area of the contact. The difference between the resistances of the stable states $\Delta R = R_{on} - R_{off}$ is then

$$
\Delta R = 2(k_2 - k_1)\frac{\sinh(p\,r/2)\sinh(p[1 - r]/2)}{\sinh(p/2)}.
$$
(4.4)

It is only non-zero if the contacts are different ($k_1 \neq k_2$), otherwise the hysteresis loop of the memristor has the characteristic 'table with legs' type [14,29], which is beyond the scope of the present consideration.

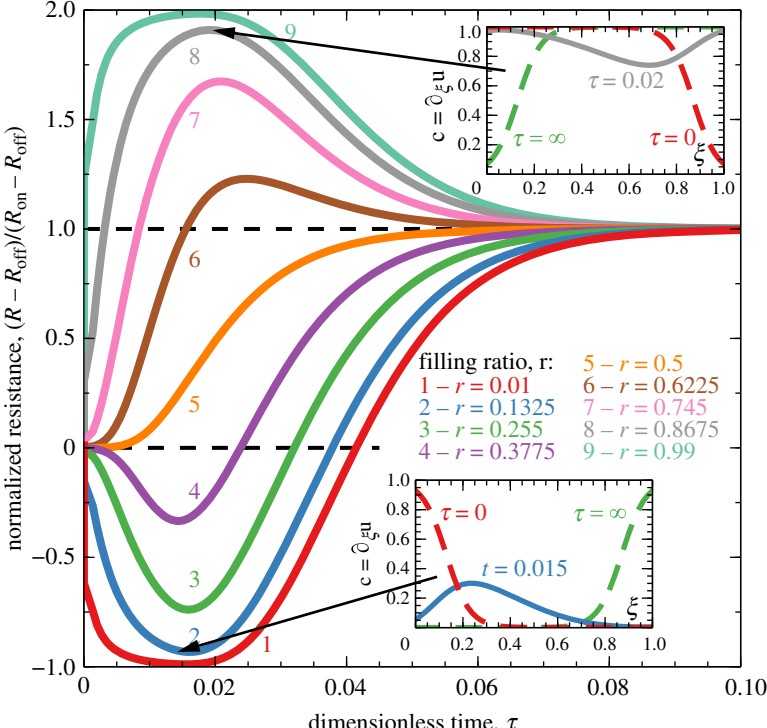

**Figure 2.** Time evolution of the normalized resistance, computed from (2.6), (3.2) and (4.3) for different filling ratios $r$ taking the values from 0.01 to 0.99 in equal steps. The other parameters are: $p = 20$, $k_1 = 1$, $k_2 = 2$. Insets show vacancy distributions around extrema of the resistance (indicated by arrows) for two values of $r$, the dashed lines in insets correspond to their respective 'on' and 'off' configurations.

The resistance is maximized at $r = 1/2$ when $\Delta R|_{r=1/2} = \Delta R_{\max} = (k_2 - k_1)\tanh(p/4)$. When the memristor is switched, its resistance changes by $\Delta R$ from $R_{\text{off}}$ to $R_{\text{on}}$ or back. Let us now consider kinetics of this process.

Figure 2 shows the time evolution of the normalized resistance during the switching. It is interesting that $R(t)$ in the figure is strictly monotonous only for $r = 1/2$, for other values of $r$, it drops below $R_{\text{off}}$ or overshoots the value of $R_{\text{on}}$ during the switching process. This happens because either (for $r < 1/2$) the vacancies depart from the $\xi = 0$ contact and move as a soliton throughout the film before they start accumulating at $\xi = 1$; or (for $r > 1/2$) the vacancies start accumulating at $\xi = 1$ before they had time to depart from $\xi = 0$. This is a strictly nonlinear effect, and for small values of $p \ll 1$, the time evolution of the resistance is monotonous for all $r$. For larger $p$, the range of $r$ values around $r = 1/2$, corresponding to the monotonous evolution, progressively shrinks. For applications, a large value of $\Delta R$ is desirable, which, as follows from (4.4), implies both large value of $p$ and the optimal filling $r = 1/2$. In this case, checking the monotonicity of the resistance relaxation under the applied electric current can be a useful tool for optimizing the filling ratio. It can indicate, based on measurements of a single sample, whether its filling ratio is above or below the optimal value.

Evolution of the resistance under the applied current is, basically, a relaxation process towards the equilibrium configuration $c_{\text{on}}$ (or $c_{\text{off}}$ for $p < 0$). It is typical that such processes approach equilibrium according to the exponential law $\propto e^{-\tau/\tau_R}$, where $\tau_R$ is the relaxation time. This is also the case for the present model. Figure 3 shows the logarithmic derivative of time evolution of the resistance. At large times, these curves become horizontal, which means that relaxation is exponential, their limiting value at $t \to \infty$ is equal to $-1/\tau_R$. It is also worth noting that memristors with optimal filling $r = 1/2$ are the first to reach exponential relaxation regime.

It is not difficult to compute the relaxation time analytically from (2.6) and (4.3) by neglecting all but the first terms in the Fourier series (as they are exponentially small, compared to the first). This gives

$$\tau_R = \frac{4}{p^2 + 4\pi^2}, \tag{4.5}$$

which is independent of $r$ and the initial distribution of vacancies. Figure 3 shows that logarithmic derivative of the resistance for different values of $p$, the filling ratio $r$ and a particular initial

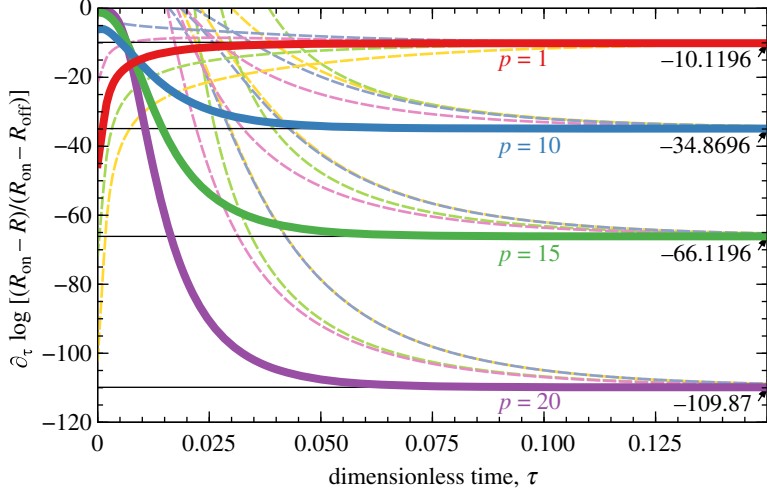

**Figure 3.** Logarithmic time derivative of the normalized resistance during the switching process for $r = 1/2$ (solid lines) and different values of $p$. The numbers on the right show the limiting values at $\tau \to \infty$ computed from (4.5). The dashed lines correspond to $r = 0.1, 0.3, 0.7, 0.9$ for each value of $p$; they illustrate that the limit is independent of $r$.

distribution of the vacancies relaxes exactly to the values $-1/\tau_R$, with $\tau_R$ given by (4.5). This simple formula contains two key characteristics of the memristor: at $p > 0$, it gives an estimate of the duration of the current pulse, necessary to switch the memristor into 'on' (or 'off') state; at $p = 0$, it gives an estimate of the memristor state lifetime $\tau_0 = 1/\pi^2$ under the influence of thermal fluctuations. Practical considerations may require to introduce factors before these times, e.g. extending the current pulse duration to be several times $\tau_R$ to ensure that bit is completely written, or consider the retention time to be several times less than $\tau_R$ to ensure that bit can still be read reliably. Nevertheless, $\tau_R$ is a convenient simple estimate for both these times.

In seconds, the relaxation time (4.5) is equal to

$$t_R = \frac{d^2}{D} \tau_R = \frac{1}{4D} \frac{d^2}{\pi^2 + (d^2/a^2) \sinh^2 \frac{qa\rho_0 I}{k_B T}}$$

$$\approx \frac{1}{4D} \frac{d^2}{\pi^2 + (dq\rho_0 I)^2/(k_B T)^2}, \tag{4.6}$$

where the last approximate equality assumes $a \ll k_B T/(q\rho_0 I)$. It has an interesting feature, which is entirely due to the nonlinear term in (2.4): at zero driving current, the relaxation time (bit lifetime in this case) scales quadratically with film thickness $d$, but at large driving current, it (bit writing time in this case) saturates for large $d$. It means that increasing the thickness (memristor length) to greater than or equal to approximately $\pi a/\sinh[(qa\rho_0 I)/(k_B T)]$ will increase the bit lifetime quadratically, but the writing time will not be significantly increased. This suggests that vacancy-based memristors in strongly nonlinear regime (with large $p$) may have promising applications for long-term information storage.

Let us also remark on the case when local resistivity $\rho_0 = \rho_{00}(1 + \gamma c)$ is weakly $\gamma \ll 1$ dependent on the vacancy concentration $c$. In the first order of perturbation theory over $\gamma$, this only leads to rescaling of the constant $R_0$ by $1 + \gamma r$ in (4.3). There is no further impact on the evolution of the resistance because the total number of vacancies $r = const$ in a contact with closed boundaries.

The main limitation of the present consideration is that it does not take into account Joule heating of the film due to the applied current and phase changes in the material. From (4.6), it follows that the temperature increase reduces the value of $p$ and effectively counteracts the effect of writing current. Thus, in practical applications, the maximum achievable value of $p$ is limited. This limit can be controlled by the selection of the memristor's material.

## 5. Conclusion

We have considered and solved exactly a simple analytical vacancy-migration model for a memristor. Its kinetics is governed by nonlinear Burger's equation with conserved number of vacancies and no vacancy

current at the boundary. There are two substantially nonlinear effects: non-monotonous relaxation of resistance under the applied current (which can be used for optimizing the initial number of vacancies in the memristor) and the saturation of the memristor switching time for increasing film thickness (which decouples the bit writing time from the bit lifetime). We hope that both these effects can be useful in the development of memristor-based memory applications and that the analytical solution (2.6) can become a benchmark for more complex memristor simulations.

Data accessibility. All data related to this theoretical research are the analytical expressions contained in the manuscript.
Authors' contributions. I.V.B. conceived the work and formulated the problem, K.L.M. did the initial solution of Burgers' equation, both authors interpreted the results and wrote the manuscript together.
Competing interests. The authors declare no competing interests.
Funding. K.L.M. acknowledges the support of the Russian Science Foundation under the project no. RSF 21-11-00325.
Acknowledgements. The authors thank Prof. V. E. Zakharov for his lectures at 'Kourovka-XXXVIII' winter school, which inspired us to look for analytical solution of the present very interesting and practical problem.

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
