## [Peer Review File · Royal Society Open Science]

Review History

RSOS-210677.R0 (Original submission)

Review form: Reviewer 1

Is the manuscript scientifically sound in its present form?

Yes

Are the interpretations and conclusions justified by the results?

Yes

Is the language acceptable?

Yes

Do you have any ethical concerns with this paper?

No

Have you any concerns about statistical analyses in this paper?

No

Recommendation?

Accept with minor revision (please list in comments)

Comments to the Author(s)

The paper's main contribution seems to be an analytical solution to the one-dimensional ionic drift-diffusion model that is representative of the resistive switching devices (i.e. memristors). Such a model could provide some useful insight for the memristor operation. The paper is well written and easy to read. I have only a few minor comments, which I list below roughly in the order of importance.

- The main weakness of the paper is that it is not solving for coupled electronic-ionic transport, and e.g. that the electric field across the bulk of the device is assumed to be constant. This is a simplification of the device's operation since the changes in the distribution of charged mobile ions and/or changes in the interface conductance should impact the electric field in the bulk of the device. Could authors provide more justification for such assumptions, e.g. discuss some special cases when such assumptions would be acceptable?
- The paper's focus is somewhat similar to that of Refs. r1, r2, r3 (see below) so it would be great to have some discussion on how the paper's results are different.
- Page 4, first paragraph: "It is interesting to know that the external force (electric current) enters the equation (2.4) as a coefficient before the nonlinear term, but not as a separate term in the right hand side". Could authors clarify this sentence and, in particular, explain why it is interesting?
- Also, I would rather use "electric field" instead of "electrical current" in the sentence above and also later on page 4 in the last paragraph to avoid confusion with ion motion due to electromigration.

[r1] D.B. Strukov et al, Coupled ionic and electronic transport model of thin-film semiconductor memristive behavior, *Small*, vol. 5, pp. 1058-1063, 2009

[r2] M. Noman et al, Computational investigations into the operating window for memristive devices based on homogeneous ionic motion, *Applied Physics A*, vol. 102, pp. 877-878, 2011.

[r3] D.B. Strukov et al. An ionic bottle for high-speed, long-retention memristive device, *Applied Physics A*, vol. 102, pp. 1033-1036, 2011.

Review form: Reviewer 2

Is the manuscript scientifically sound in its present form?

Yes

Are the interpretations and conclusions justified by the results?

Yes

Is the language acceptable?

Yes

Do you have any ethical concerns with this paper?

No

Have you any concerns about statistical analyses in this paper?

No

Recommendation?

Accept with minor revision (please list in comments)

Comments to the Author(s)

The manuscript presents an analytical way to describe memristive phenomena in resistive switching materials. One cannot verify all the details surrounding the mathematical derivation the authors present but their results are very interesting and sound. There are many memristive models in the literature that address the problem with simulations, therefore, it is interesting to have an analytical approach that can be contrasted with simulations. The manuscript is well presented and I am glad to recommend it for publication (with minor revisions). My only remarks are that the authors could perhaps provide a supplemental material or some more details concerning the technical parts of solving those equations numerically. Or perhaps, just include an extra section in the manuscript that details about their main numerical methods used to solve the equations. Any numerical tolerances to consider? Thresholds? Truncation of series? Integration methods? Did the authors test the approach in other case studies beyond those presented in the manuscript? These would help the community to try to reproduce their results and perhaps extend the description to more complex cases as the authors suggest. I also would like to request if the authors could, please, explain better the result of Figure 3 that is not so clear to me.

Decision letter (RSOS-210677.R0)

Dear Dr Boylo

On behalf of the Editors, we are pleased to inform you that your Manuscript RSOS-210677 "Nonlinear effects in memristors with mobile vacancies" has been accepted for publication in Royal Society Open Science subject to minor revision in accordance with the referees' reports. Please find the referees' comments along with any feedback from the Editors below my signature.

Please submit your revised manuscript and required files (see below) no later than 7 days from today's (ie 28-Sep-2021) date. Note: the ScholarOne system will 'lock' if submission of the revision is attempted 7 or more days after the deadline. If you do not think you will be able to meet this deadline please contact the editorial office immediately.

on behalf of Professor Tim Rogers (Associate Editor) and Miles Padgett (Subject Editor)
openscience@royalsociety.org

Reviewer comments to Author:

Reviewer: 1

Comments to the Author(s)

The paper's main contribution seems to be an analytical solution to the one-dimensional ionic drift-diffusion model that is representative of the resistive switching devices (i.e. memristors). Such a model could provide some useful insight for the memristor operation. The paper is well written and easy to read. I have only a few minor comments, which I list below roughly in the order of importance.

- The main weakness of the paper is that it is not solving for coupled electronic-ionic transport, and e.g. that the electric field across the bulk of the device is assumed to be constant. This is a simplification of the device's operation since the changes in the distribution of charged mobile ions and/or changes in the interface conductance should impact the electric field in the bulk of the device. Could authors provide more justification for such assumptions, e.g. discuss some special cases when such assumptions would be acceptable?

- The paper's focus is somewhat similar to that of Refs. r1, r2, r3 (see below) so it would be great to have some discussion on how the paper's results are different.

- Page 4, first paragraph: "It is interesting to know that the external force (electric current) enters the equation (2.4) as a coefficient before the nonlinear term, but not as a separate term in the right hand side". Could authors clarify this sentence and, in particular, explain why it is interesting?

- Also, I would rather use "electric field" instead of "electrical current" in the sentence above and also later on page 4 in the last paragraph to avoid confusion with ion motion due to electromigration.

[r1] D.B. Strukov et al, Coupled ionic and electronic transport model of thin-film semiconductor memristive behavior, *Small*, vol. 5, pp. 1058-1063, 2009

[r2] M. Noman et al, Computational investigations into the operating window for memristive devices based on homogeneous ionic motion, *Applied Physics A*, vol. 102, pp. 877-878, 2011.

[r3] D.B. Strukov et al. An ionic bottle for high-speed, long-retention memristive device, *Applied Physics A*, vol. 102, pp. 1033-1036, 2011.

Reviewer: 2

Comments to the Author(s)

The manuscript presents an analytical way to describe memristive phenomena in resistive switching materials. One cannot verify all the details surrounding the mathematical derivation the authors present but their results are very interesting and sound. There are many memristive models in the literature that address the problem with simulations, therefore, it is interesting to have an analytical approach that can be contrasted with simulations. The manuscript is well

presented and I am glad to recommend it for publication (with minor revisions). My only remarks are that the authors could perhaps provide a supplemental material or some more details concerning the technical parts of solving those equations numerically. Or perhaps, just include an extra section in the manuscript that details about their main numerical methods used to solve the equations. Any numerical tolerances to consider? Thresholds? Truncation of series? Integration methods? Did the authors test the approach in other case studies beyond those presented in the manuscript? These would help the community to try to reproduce their results and perhaps extend the description to more complex cases as the authors suggest. I also would like to request if the authors could, please, explain better the result of Figure 3 that is not so clear to me.

===PREPARING YOUR MANUSCRIPT===

===PREPARING YOUR REVISION IN SCHOLARONE===

Author's Response to Decision Letter for (RSOS-210677.R0)

See Appendix A.

Decision letter (RSOS-210677.R1)

Dear Dr Boylo,

I am pleased to inform you that your manuscript entitled "Nonlinear effects in memristors with mobile vacancies" is now accepted for publication in Royal Society Open Science.

on behalf of Professor Tim Rogers (Associate Editor) and Miles Padgett (Subject Editor)
openscience@royalsociety.org

Appendix A

Response to Reviewers – RSOS-210677

We thank the reviewers for their work in reviewing our manuscript and their positive evaluation of our work as a whole. We have carefully read the reports and revised the manuscript to address all the reviewers' comments and concerns. The changes were minor, yet they indeed help to improve the quality of the manuscript. Here is the list of our responses and the resulting corrections:

I. Comments and questions of the Reviewer 1:

1. *“The main weakness of the paper is that it is not solving for coupled electronic-ionic transport, and e.g. that the electric field across the bulk of the device is assumed to be constant. This is a simplification of the device's operation since the changes in the distribution of charged mobile ions and/or changes in the interface conductance should impact the electric field in the bulk of the device. Could authors provide more justification for such assumptions, e.g. discuss some special cases when such assumptions would be acceptable?”*

RESPONSE:

We agree with the Reviewer that our model (as any other) is limited and there is a set of phenomena, which is not taken into account. Specifically, we consider the type of the memristors in which the change of the resistance is an interfacial effect. This limitation (that resistivity ρ_0 is constant and, consequently, the driving electric field is also constant throughout the bulk of our memristor) is crucial for us to be able to solve the model exactly and analytically. To convey this point more clearly, we have added the following text to the end of the paragraph, containing the equation (4.1): “This resistance model corresponds to the case (like in some transition metal oxides [ourREF14,r2]), when the bulk resistance change is negligible compared to the interfacial one. The opposite situation may also take place [r1,r3], but is beyond the scope of the present analytical treatment.”

2. *“The paper's focus is somewhat similar to that of Refs. r1, r2, r3 (see below) so it would be great to have some discussion on how the paper's results are different.*

[r1] D.B. Strukov et al, Coupled ionic and electronic transport model of thin-film semiconductor memristive behavior, Small, vol. 5, pp. 1058-1063, 2009

[r2] M. Noman et al, Computational investigations into the operating window for memristive devices based on homogeneous ionic motion, Applied Physics A, vol. 102, pp. 877-878, 2011.

[r3] D.B. Strukov et al. An ionic bottle for high-speed, long-retention memristive device, Applied Physics A, vol. 102, pp. 1033-1036, 2011.
”

RESPONSE: We thank the Reviewer for the references ! We have already pinpointed one crucial difference with [r1,r3] in the answer to the previous comment. The [r2] paper is an interesting numerical simulation paper we have now added to the list of other recent modeling works cited in the last paragraph of the first section. The main difference from [r2], which also studies memristors with interfacial resistance, is that our model admits an analytical solution, which we were lucky to find.

3. *“- Page 4, first paragraph: “It is interesting to know that the external force (electric current) enters the equation (2.4) as a coefficient before the nonlinear term, but not as a separate term in the right hand side”. Could authors clarify this sentence and, in particular, explain why it is interesting?”*

RESPONSE: In the classical equations of mathematical physics (like the wave equation of the linear diffusion/heat equation) the driving force usually appears on the right hand side and makes the equation non-homogeneous (meaning that trivial solution, equal to zero everywhere, is not a solution of the equation anymore). Our case is different. This is by no means unique in physics, just a little unusual. In magnetism (an example, probably, too far-fetched to be mentioned in the manuscript) it is, for example, quite unusual, that the spin-polarized current enters the equations of motion of magnetic moments not as a driving force, but as a negative dissipation. To specify more precisely what we mean, we have changed the relevant sentence into: “It is interesting that the external force (electric field $\rho_0 I$) enters the equation (2.4) via a coefficient before the nonlinear term, but not as a separate term in the right hand side like in many classical equations of mathematical physics”.

4. *“Also, I would rather use “electric field” instead of “electrical current” in the sentence above and also later on page 4 in the last paragraph to avoid confusion with ion motion due to electromigration.”*

RESPONSE: In the considered case of constant ρ_0 we can use the electric current and the electric field, acting on the vacancies, interchangeably (the field and the current are related by the Ohm law with the constant resistivity ρ_0). But to avoid the potential confusion referee mentioned we have replaced “electrical current” by the “electric field” after the Eq. (2.4) and the “current” by the “field” on page 4 (after the Eq. 3.1) .

II. Comments and questions of the Reviewer 2:

1. *“My only remarks are that the authors could perhaps provide a supplemental material or some more details concerning the technical parts of solving those equations numerically. Or perhaps, just include an extra section in the manuscript that details about their main numerical methods used to solve the equations. Any numerical tolerances to consider? Thresholds? Truncation of series? Integration methods? Did the authors test the approach in other case studies beyond those presented in the manuscript? These would help the community to try to reproduce their results and perhaps extend the description to more complex cases as the authors suggest.”*

RESPONSE: Our work is analytical and the validity of our results can mostly be established by a simple substitution (e.g. the solution (2.6) into the equation (2.4) and its boundary conditions). This includes the dashed curves in Fig. 3, which are not the result of a numerical simulation, but simply a rendering (via the analytically computed logarithmic derivative of the resistance) of the exact time evolution of an initial state (3.2) according to the equation (2.6). But the referee is right that we have missed a detail of the numerical computation. The solution (2.6) is only exact as it is written, i.e. with infinite summation. In practice these Fourier-type series, while they provably converge, do not converge very fast. We have added the sentence to the last paragraph of the Section 3: “We have used a finite number (3000) terms in the infinite series (2.6), which is sufficient for plotting the

subsequent Figures.”

2. *“I also would like to request if the authors could, please, explain better the result of Figure 3 that is not so clear to me.”*

RESPONSE: The Figure 3 demonstrates that the logarithmic derivative of the resistance during the resistance switching relaxes (as time passes) to a value $-1/\tau_R$, with τ_R given by a simple (initial condition independent) formula (4.5). We have added the sentence after the Eq. (4.5) to clarify this role of Fig. 3: “One can directly see in Fig. 3 that logarithmic derivative of the resistance for different values of p , the filling ratio r and a particular initial distribution of the vacancies, relaxes exactly to the values $-1/\tau_R$, with τ_R given by (4.5).”

We thank again the Reviewers for their very useful remarks.

With the best regards,

Dr. I.V. Boylo

Dr. K.L. Metlov